# Are Protective Activities and Limitations in Practical Skills of Daily Living Associated with the Cognitive Performance of People with Mild Cognitive Impairment? Baseline Results from the BrainFit-Nutrition Study

**DOI:** 10.3390/nu15163519

**Published:** 2023-08-10

**Authors:** Petra Scheerbaum, Elmar Graessel, Sophia Boesl, Etienne Hanslian, Christian S. Kessler, Julia-Sophia Scheuermann

**Affiliations:** 1Center for Health Service Research in Medicine, Department of Psychiatry and Psychotherapy, Uniklinikum Erlangen, Friedrich-Alexander-Universität Erlangen-Nürnberg (FAU), 91054 Erlangen, Germany; petra.scheerbaum@uk-erlangen.de (P.S.);; 2Institute of Social Medicine, Epidemiology and Health Economics, Charité—Universitätsmedizin Berlin, Corporate Member of Freie Universität Berlin and Humboldt-Universität zu Berlin, 10117 Berlin, Germany; 3Department of Internal and Integrative Medicine, Immanuel Krankenhaus Berlin, 14109 Berlin, Germany

**Keywords:** mild cognitive impairment, dementia, lifestyle medicine, chronic diseases, integrative medicine

## Abstract

Limitations in daily living have not yet been described adequately for mild cognitive impairment (MCI). In this study, we investigated first, time spent on protective activities (social, mental, and physical) and second, limitations in practical skills of daily living, both for people with MCI. We used baseline data from 270 individuals who participated in the randomized controlled trial BrainFit-Nutrition. The Montreal Cognitive Assessment (MoCA) was used to identify people with MCI. Participants were asked how much time they spent engaged in social, mental, and physical activities each week. Furthermore, the Bayer-ADL scale was used to quantify deficits in activities of daily living (ADLs). Regarding protection, the number of hours spent engaged in the three activity areas was significantly correlated with the cognitive performance in people with MCI. Social activities were positively associated with current cognitive performance. Concerning the limitations in practical skills of daily living, older and more cognitively impaired individuals were affected. Memory and orientation appear to be among the first practical skills of daily living that become impaired in people with MCI. Treatment recommendations for people with MCI include an increase in social, mental, and physical activities as well as the promotion of a healthy lifestyle.

## 1. Introduction

As a result of demographic changes, society is increasingly burdened by the rising prevalence of dementia. At the same time, the age of onset of dementia is continuously decreasing [1]. At least 50 million people worldwide live with this diagnosis [2]. The prevalence of mild cognitive impairment (MCI), which often precedes dementia [3,4], also increases with age; between 6.7% of 60- to 64-year-olds and 25.5% of 80- to 84-year-olds are concerned [5]. Compared with cognitively normal people, individuals with MCI have a higher risk of progressing to dementia [4,6].

Various activities, such as physical exercise, cognitive training, as well as maintaining social contacts [7,8,9,10], can improve cognitive performance for cognitively impaired individuals. Herewith, social, mental, and physical activities can protect people with MCI against cognitive decline.

Concerning social activities, maintaining social contacts seems to protect people with MCI against cognitive decline [10,11]. Other authors found that an increase in social engagement can reduce the risk of conversion to dementia [8] and identified a positive relationship between social activities and cognitive status [12].

Regarding mental activities, engaging in mentally demanding tasks has been found to protect people against cognitive decline [13,14]. Consequently, training various cognitive skills several times a week can help preserve the cognitive abilities [7] of cognitively healthy people as well as people with cognitive impairment and people with dementia [15]. Early initiation of cognitive training has been demonstrated to be particularly effective, especially with individualized computerized programs [16].

According to global recommendations on physical activity for health from the World Health Organization (WHO) [17], maintaining a regular activity level in the elderly population (65 years and older) is associated with several health benefits. A non-adherence to the recommendations may be associated with more frequent occurrences of diseases, such as high blood pressure, stroke, or type 2 diabetes. These diseases have already been identified in a review as risk factors for dementia [18]. Furthermore, a higher level of physical activity has been reported to protect people from the effects of dementia, especially for those with Alzheimer’s disease [19]. For MCI, the benefits of physical activity concerning cognitive performance have also been confirmed [9].

In the dementia stage, limitations in ADL can occur [20]. The basic ADLs include self-maintenance tasks, such as personal hygiene, dressing, or eating [21,22]. Furthermore, there is also evidence of limitations in instrumental activities of daily living for people with dementia [23]. Instrumental ADLs refer, for example, to cooking, laundry, and shopping [22]. Certain limitations in instrumental ADLs can already be observed in the MCI stage [24,25]. 

As there is still no effective drug intervention for reducing the risk of the transition from MCI to dementia [5,26], the implementation of non-pharmacological therapies should be prioritized [27,28,29]. Therefore, treatment recommendations for MCI include an increase in social, mental, and physical activities as well as the promotion of a healthy lifestyle in the early stages [30] to slow down the condition from progressing to dementia. In the previous literature, limitations in practical skills of daily living—including both basic and instrumental activities of daily living—have been described as the primary components of the dementia syndrome [26,31]. However, it remains unclear how exactly the activity level and limitations in practical skills of daily living are characterized in people with MCI. 

Therefore, the goals of the present secondary data analysis were twofold: first, to determine the protective character of the social, mental, and physical activity concerning the cognitive performance of people with MCI, and second, to identify to which extent the limitations in practical skills of daily living are correlated to the cognitive performance of people with MCI.

## 2. Materials and Methods

### 2.1. Design and Sample

We used baseline data from the randomized controlled trial BrainFit-Nutrition for the analysis. The study design has already been published as a study protocol [32]. The design comprises a prospective 2 × 2 randomized controlled trial. The intervention study for people with MCI consists of two intervention arms: nutritional counseling and computerized cognitive training (CCT). In both arms, we developed an active intervention specialized for people with MCI and an active control measure. Concerning nutrition, a whole-food, plant-based diet (WFPB) was the specialized intervention. For the CCT, the specialized intervention was the individualized self-learning system. To complement the interventions, two kinds of active control measures were used. For nutrition, it was a healthy omnivorous diet based on the current guidelines of the German Nutrition Society (Deutsche Gesellschaft für Ernährung; DGE). Concerning the CCT, basic tasks that include path targeting, simple strategies, and long-term memory were developed. The main objectives of the BrainFit-Nutrition study were to test the efficacy of the two interventions compared with the active control interventions. In the present study, based on the described study design, a secondary data analysis is performed with the baseline data.

More than 1100 people across Germany aged 60 and older were screened for MCI between January and September 2022. To be able to offer participation to all residents of Germany, the study was conducted completely remotely. Exclusion criteria for participation were the following: completely blind or deaf; no personal computer, laptop, or tablet; normal cognition, Montreal Cognitive Assessment [33] (MoCA) score > 24; Dementia, Mini-Mental State Examination [34] (MMSE) score < 24; Depression, Patient Health Questionnaire 9 [35] (PHQ-9) score ≥ 12; diagnosis of another disease that causes cognitive impairment, e.g., psychosis, morbus Parkinson, or multiple strokes [32]. Participants were randomly allocated to one of four groups and received a combination of online nutritional counseling and a CCT. The allocation has been stratified by age, gender, and baseline MoCA score. The primary inclusion criterion of the psychometric determination of MCI as a MoCA score ≤ 24 points and a MMSE score ≥ 24 points identified 326 individuals as having MCI; of these, 271 individuals with MCI gave their written consent to participate in the study and were randomized. A total of *N* = 270 completed the baseline survey in a telephone interview. All procedures were approved by the Ethics Committee of the medical faculty of the Friedrich-Alexander-Universität Erlangen-Nürnberg (Ref.: 21-318_1-B). A prospective registration was carried out by the International Standard Randomized Controlled Trial Number Registry (ISRCTN 10560738). 

### 2.2. Instruments

To determine the current cognitive performance of people with MCI, the standardized psychometric test MoCA [33] was conducted. The sum score ranges between 0 and 30 points. Further, we collected activity-related, everyday practical, and socio-demographic data in a standardized questionnaire. The socio-demographic data comprised age, gender, employment status, living situation, and highest educational level. For age, we have created age quartiles (60 to 64 years, 65 to 69 years, 70 to 74 years, 75 years and older). Employment status was measured by asking whether the person was still working (yes/no). For the living situation, we identified whether the person was living alone (yes/no). Education was measured according to the International Standard Classification of Education (ISCED) [36]. Moreover, we included basic information concerning the physical health status, such as weight and height, to measure BMI, hypertension (yes/no), and hypercholesterinemia (yes/no). 

As for the first research question, the daily time spent engaged in social, mental, and physical activities was recorded to check the potentially protective character concerning cognitive performance. The participants could report a weekly activity amount between 0 and 70 h per week, since higher values were not considered plausible. 

As for the second research question, the Bayer Activities of Daily Living scale (Bayer-ADL scale) [37] was administered to quantify the deficits in practical skills of daily living. Initially used as a third-party assessment, the Bayer-ADL scale can also be used as a self-assessment tool. As individuals with MCI are able to answer for themselves, the self-assessed Bayer-ADL scale was used in this study. The Bayer-ADL scale consists of 25 items regarding the subscales introductory or general activities of daily living (ADLs) competencies (2 items), specific tasks of daily living (12 items), short- or long-term memory (3 items), orientation (3 items), and cognitive skills (5 items) [38,39]. Example items from the subscales are: “Do you have difficulties taking care of yourself?” (General coping); “Do you have difficulties preparing food?” (Specific tasks); “Do you have difficulties observing important dates or events?” (Memory); “Do you have difficulties going for a walk without getting lost?” (Orientation); and “Do you have difficulties doing two things at the same time?” (Cognitive skills). The items were answered on a 10-point scale ranging from 1 (never) to 10 (always); for a non-applicable item, the answer option “Not applicable” was available. A sum score for the practical skills of daily living was computed in addition to the subscale scores.

### 2.3. Statistical Analysis

The analysis was conducted on *N* = 270 participants. With the exception of the income variable (7 [2.6%] missing values), there were no missing values. Pearson correlations were performed to examine the association of activity levels and limitations of practical skills of daily living with the current cognitive performance in the MoCA. Subgroup analyses were conducted in terms of gender, age, and cognitive impairment. In order to determine whether the associations were a function of cognitive impairment, the MoCA score was dichotomized at the median. To consider group differences, we conducted *t*-tests for independent samples or analyses of variance (ANOVA). An accumulation of the alpha error due to multiple testing was corrected with the Benjamini–Hochberg procedure [40]. In a sensitivity analysis, the relevance of the bivariate significant associations was assessed in a multivariable analysis using multiple linear regressions. The analyses (*p* < 0.05) were carried out with IBM SPSS Statistics 28.0.

## 3. Results

The mean age was 70.8 years (*SD* = 6.9). About half of the sample was female (52.2%), almost one-third lived alone (29.6%), three-fifths had a high level of education (58.9%), and the majority of the participants were no longer working (71.9%). The mean body mass index (BMI) was within normal range (BMI = 25.5). Almost half of our participants suffered from hypertension or hypercholesterinemia. Beyond that, the other risk factors, for instance, diabetes, had a low prevalence in the sample (<15%). Hence, we did not include health status in further analysis. The sample characteristics are given in Table 1.

### 3.1. Activity Level of Individuals with MCI by Sex, Age, and Cognitive Performance

The total amount of time that individuals with MCI spent on social, mental, and physical activities were significantly but moderately correlated with the current cognitive performance, as assessed with the MoCA sum score (*r* = 0.134, *p* = 0.032). The time spent engaged in various aspects of daily activities was not significantly correlated with the current cognitive performance, but for both mental (*r* = 0.108, *p* = 0.076) and social activities (*r* = 0.111, *p* = 0.068), a statistical trend could be observed. On the contrary, physical activities had a non-significant correlation of *r* = 0.056 (*p* = 0.356). In the multivariable model (*F* [1, 268] = 4.93, *p* = 0.027, *R*^2^_corrected_ = 0.014), overall daily activity was a significant factor for current cognitive performance (β = 0.134, *p* = 0.027).

Women reported higher levels of social activities (*t* = −2.220, *df* = 268, *p* = 0.027) and a greater total time spent engaged in daily activities (*t* = −2.154, *df* = 268, *p* = 0.032) than men. There were no gender differences for mental (*t* = −1.750, *df* = 268, *p* = 0.081) or physical activities (*t* = −0.481, *df* = 268, *p* = 0.631) in the present sample.

The sex-specific subgroup analysis did not reveal any associations with the current cognitive performance. In the sub-population of women (*n* = 141), the following non-significant correlations were obtained: the correlation between the current cognitive performance and total activities was *r* = 0.094 (*p* = 0.267). For the activity dimensions, the correlation between the current cognitive performance and mental activities was *r* = 0.055 (*p* = 0.516). The correlation between the current cognitive performance and physical activities was *r* = 0.089 (*p* = 0.295), and the correlation between the current cognitive performance and social activities was *r* = 0.059 (*p* = 0.487).

In the subpopulation of men (*n* = 129), the following non-significant associations were found: the association between the current cognitive performance and total activities was *r* = 0.134 (*p* = 0.131). For the activity dimensions, the current cognitive performance showed the following correlations: mental activities, *r* = 0.132 (*p* = 0.135); physical activities, *r* = 0.013 (*p* = 0.880); and social activities, *r* = 0.119 (*p* = 0.180).

There were no differences in activity levels between the age quartiles: they did not differ with respect to overall activities (*F*[3, 266] = 0.228; *p* = 0.877) or the dimensions of mental (*F*[3, 266] = 0.284; *p* = 0.837), physical (*F*[3, 266] = 0.545; *p* = 0.652), or social activities (*F*[3, 266] = 0.277; *p* = 0.842). The age-related subgroup analysis revealed relevant results only for individuals with MCI in the 70- to 75-year-old age group: the time spent on mental activities (*r* = 0.285, *p* = 0.028) and the total amount of time spent on all activities (*r* = 0.264, *p* = 0.034) were significantly but moderately correlated with the current cognitive performance. The amount of time spent engaged in physical activities (*r* = 0.044, *p* = 0.727) or social activities (*r* = 0.126, *p* = 0.313) were not significantly associated with the current cognitive performance in this age group. The multivariable model for this age group (*F*[2, 63] = 3.06, *p* = 0.054) reached no significance. For the other three age quartiles, there were no significant associations with the current cognitive performance score for either the total or the activity dimensions (*p* > 0.05).

The median dichotomization of the MoCA resulted in a split at 23. There were no significant differences in activity levels between individuals with higher MoCA scores (MoCA ≥ 23; *n* = 142) and individuals with lower MoCA scores (MoCA < 23; *n* = 128). Individuals did not differ in terms of total activity level (*t* = −1.505, *df* = 268, *p* = 0.133) or in terms of the dimensions mental activities (*t* = −1.585, *df* = 268, *p* = 0.114), physical activities (*t* = −0.779, *df* = 268, *p* = 0.437), or social activities (*t* = −0.388, *df* = 268, *p* = 0.698).

The cognition-related subgroup analysis revealed a significant correlation only for the individuals with lower MoCA scores: For these individuals, the time spent engaged in social activities was significantly correlated with the current cognitive performance (*r* = 0.207, *p* = 0.028). The total amount of time spent engaged in any activities (*r* = 0.105, *p* = 0.238) and the two dimensions of mental activities (*r* = 0.015, *p* = 0.867) and physical activities (*r* = 0.049, *p* = 0.585) did not show significant correlations with the current cognitive performance. In the multivariable model (*F*[1, 126] = 5.63, *p* = 0.019, *R*^2^_corrected_ = 0.035), social activity was a significant factor for current cognitive performance (β = 0.207, *p* = 0.019).

For the subgroup of individuals with higher MoCA scores, there were no significant correlations with the current cognitive performance for the total amount of activities (*r* = 0.131, *p* = 0.120) or for the dimensions mental (*r* = 0.148, *p* = 0.079), physical (*r* = 0.006, *p* = 0.939), and social activities (*r* = 0.090, *p* = 0.289).

### 3.2. Limitations in Practical Skills of Daily Living by Sex, Age, and Cognitive Performance

Most of the individuals with MCI (80–95%) did not experience any subjective limitations in the sum score for limitations in practical skills of daily living as assessed with the Bayer-ADL scale (Table 2). The findings were the same for the subscales, with the exception of cognitive skills, where 63% reported no subjective limitations. The memory (*r* = −0.243, *p* = 0.012), orientation (*r* = −0.175, *p* = 0.016), and cognitive skills (*r* = −0.225, *p* = 0.012) subscales and the sum score of all everyday limitations (*r* = −0.206, *p* = 0.028) were significantly and slightly negatively correlated with current cognitive performance. The subscales of general ADL competencies (*r* = −0.073, *p* = 0.232) and specific tasks (*r* = −0.103, *p* = 0.091) did not show significant associations with the current cognitive performance. In the multivariable model (*F*[3, 266] = 6.25, *p* < 0.001, *R*^2^_corrected_ = 0.055), only memory was a significant factor for current cognitive performance (β = −0.239, *p* = 0.007). Orientation (β = −0.114, *p* = 0.217) and the sum score of all everyday limitations (β = 0.056, *p* = 0.644) did not reach significance.

Regarding memory, men with MCI reported limitations more often than women (*F*[1, 268] = 4.54, *p* = 0.034, η^2^ = 0.02), but for orientation, women reported limitations more often than men (*F*[1, 239.1] = 6.17, *p* = 0.028, η^2^ = 0.02). There were no relevant gender differences with respect to the dimensions of general ADL competencies (*F*[1, 268] = 0.05, *p* = 0.824), specific tasks (*F*[1, 268] = 0.12, *p* = 0.913), and cognitive skills (*F*[1, 268] = 0.66, *p* = 0.416), or for the sum score of all everyday limitations (*F*[1, 268] = 0.02, *p* = 0.888).

The gender-related subgroup analysis revealed a significant but small correlation between orientation ability and current cognitive performance (*r* = −0.194, *p* = 0.021) for the subpopulation of female individuals with MCI. There were no relevant correlations with participants’ current cognitive performance for the other subdimensions of general ADL competencies (*r* = −0.011, *p* = 0.897), specific tasks (*r* = −0.007, *p* = 0.933), short- and long-term memory (*r* = −0.069, *p* = 0.419), and cognitive skills (*r* = −0.091, *p* = 0.281), or for the sum score of all everyday limitations (*r* = −0.082, *p* = 0.334). Orientation was in the multivariable model for the female subpopulation (*F*[1, 139] = 5.44, *p* = 0.021, *R*^2^_corrected_ = 0.031) also a significant factor for current cognitive performance (β = −0.194, *p* = 0.021).

For the male subpopulation, the sum score and all the dimensions showed significant negative correlations with current cognitive performance: the sum score of all everyday limitations was significantly but moderately correlated with the current cognitive performance (*r* = −0.383, *p* = 0.012). For the subdimensions, the following was found: general ADL competencies had a low correlation (*r* = −0.199, *p* = 0.024), specific tasks had a moderate correlation (*r* = −0.247, *p* = 0.014), short- and long-term memory had a moderate correlation (*r* = −0.354, *p* = 0.012), orientation had a moderate correlation (*r* = −0.260, *p* = 0.013), and cognitive skills had a moderate correlation (*r* = −0.352, *p* = 0.012) with current cognitive performance. In the multivariable model (*F*[5, 123] = 4.76, *p* < 0.001, *R*^2^_corrected_ = 0.128), only memory was a significant factor for current cognitive performance in the male subpopulation (β = −0.246, *p* = 0.017).

Significant age differences were found for general coping (*F*[3, 145.1] = 4.28, *p* = 0.028, η^2^ = 0.05), and cognitive skills (*F*[3, 147.1] = 2.93, *p* = 0.016, η^2^ = 0.03) as well as for the sum score of all everyday limitations (*F*[3, 147.3] = 2.91, *p* = 0.027, η^2^ = 0.03). For these three aspects, the post hoc test showed significant differences between the youngest (60–64 years) and oldest (≥75 years) age quartiles.

There were significant differences in limitations of practical skills of daily living between individuals with higher MoCA scores (MoCA ≥ 23; *n* = 142) and individuals with lower MoCA scores (MoCA < 23; *n* = 128). People with lower MoCA scores reported more often limitations regarding the sum score of all everyday limitations (*t* = 2.032, *df* = 268, *p* = 0.043), and the sub-dimensions short- and long-term memory (*t* = 2.861, *df* = 268, *p* = 0.005), as well as orientation (*t* = 2.109, *df* = 268, *p* = 0.036), than people with higher MoCA scores. Individuals did not differ in terms of general ADL competencies (*t* = 0.080, *df* = 268, *p* = 0.937), specific tasks (*t* = 0.985, *df* = 268, *p* = 0.325), and cognitive skills (*t* = 1.952, *df* = 268, *p* = 0.052).

The cognition-related subgroup analysis revealed a significant correlation only for the individuals with lower MoCA scores: For these individuals, the total practical skills of daily living (*r* = −0.257, *p* = 0.003), as well as the sub-dimensions short- and long-term memory (*r* = −0.207, *p* = 0.019), and cognitive skills (*r* = −0.315, *p* < 0.001) were significantly but moderately correlated with the current cognitive performance. General ADL competencies (*r* = −0.150, *p* = 0.090), specific tasks (*r* = −0.120, *p* = 0.176), and orientation (*r* = −0.159, *p* = 0.074) did not show significant correlations with the current cognitive performance. In the multivariable model (*F*[3, 124] = 3.97, *p* = 0.010, *R*^2^_corrected_ = 0.066), none of the parameters tested were found to be significant.

For the subgroup of individuals with higher MoCA scores, there were no significant correlations with the current cognitive performance for the total practical skills of daily living (*r* = −0.113, *p* = 0.181) or for the sub-dimensions of general ADL competencies (*r* = −0.125, *p* = 0.138), specific tasks (*r* = −0.008, *p* = 0.300), short- and long-term memory (*r* = −0.134, *p* = 0.113), orientation (*r* = −0.057, *p* = 0.502), and cognitive skills (*r* = −0.104, *p* = 0.219).

## 4. Discussion

### 4.1. Activity Level

The total amount of time spent engaged in social, mental, and physical activities was significantly correlated with the current cognitive performance. This association has also been revealed in the regression model. In line with the MCI therapy recommendation [30], a higher level of activity in several domains was found to be positively correlated with the current cognitive performance of individuals with MCI. However, the identified correlation was slight. Potential explanations for the small effect could be that the differentiation of current cognitive performance on the basis of MoCA scores is not fine-grained enough or because individuals with MCI provided self-assessments [25]. In contrast to the previous literature [9,42], we found no correlation between the amount of time spent engaged in physical activities and current cognitive performance. As for mental and social activities, the correlation was not significant, but a statistical trend was revealed. These findings add to the results from the previous literature regarding the association between cognitive performance and being mentally and socially active [7,8,10,11,15]. The lack of a strong correlation could possibly be explained by the self-reported amounts of time spent engaged in social, mental, or physical activities, whereby no differentiation was made with regard to intensity or difficulty.

In line with the literature, there was a positive influence of social activities on the cognitive status of the participants [12]. Women with MCI were found to invest more time in their social contacts and to be more active than men across all activities. According to an Australian study, elderly women maintain more social contacts than elderly men [43]. Thereby, the different kinds of social activities might account for the gender difference. Future research could investigate these ideas further.

For severe MCI, social activity seems to have a positive association with current cognitive performance [10,11,44]. These results were also significant in the multivariable analysis. This effect might be useful for differentiating between individuals with MCI who are more or less affected by cognitive deficits. Longitudinal studies should be conducted to test whether social activity can predict the development of cognitive performance in individuals with MCI.

Individuals with MCI who have not yet reached old age but are already in retirement seem to be able to compensate for their initial limitations. Through mental activity, they try to prevent a decline in their cognitive performance. This finding seems to be in line with the current literature, which has concluded that this age cohort tends to make more health-conscious life choices than the younger cohort and does not suffer from impairment as much as the younger ones do [45]. Due to these findings, it can be assumed that today’s younger generations will suffer from cognitive decline even more quickly when they reach an older age. Thus, it is all the more important to take preventive action to avoid an even greater burden on the healthcare system. However, it should be noted that the results were only significant in the bivariate analysis, yet not in the multivariable.

### 4.2. Basic ADLs and Instrumental ADLs

Consistent with the literature [46,47], more individuals with MCI experience subjective limitations in cognitive performance according to the Bayer-ADL scale, including orientation, memory, and cognitive skills, than in other categories of the Bayer-ADL scale. These subscales include, in particular, instrumental ADLs that were previously associated with cognitive impairment compared with basic ADLs [21,24,25]. In line with the literature, the analysis provides an indication of the presence of limitations in ADLs, particularly instrumental ADLs, for individuals with MCI [24]. The results are evident at least in the advanced stages of MCI and in the operationalization of MCI using a screening tool such as the MoCA. Because the ability to perform practical skills of daily living strongly affects health-related quality of life [48], training in the skills of daily living should be a core element of interventions in the MCI stage. In particular, individuals with MCI should receive training in instrumental ADLs, as instrumental ADLs significantly predict cognitive performance [21,49,50].

Regarding the gender difference, women reported limitations in orientation more often than men. These gender differences in orientation are in line with findings of gender differences in navigation tasks reported in a meta-analysis [51]. Furthermore, in line with the literature, men reported memory limitations more often [52]. Loprinzi et al. concluded that the differences may be due to disparities in physiology and socialization. According to Nazareth et al. [51], underlying gender roles could be partially explanatory for found differences, but they note that more research is needed.

The age-specific analysis revealed significant differences between the youngest (60–64 years) and the oldest age cohort (75 years and older). Concerning the cognitive skills as well as the sum of the practical skills, the youngest people with MCI in our study performed better than the oldest ones. As a result of the natural aging process, very elderly individuals with MCI experienced subjectively more limitations in their cognitive performance and general coping in daily life than working-age people with MCI.

The results showed a cognition-dependent difference between individuals with MCI in terms of the total practical skills of daily living, and the Bayer-ADL scales for memory and orientation. For individuals with advanced MCI, more limitations in the overall practical skills of daily living, as well as memory and cognitive skills, occurred with an increase in cognitive impairment. However, these results can be considered only as a basis for further research, as they were not confirmed in the multivariable analysis of the present data. Thereby, attention should be paid to a survey of everyday practical skills using a third-party assessment. According to Zhao et al. [53], the strength of the association between cognitive performance and IADL skills in older people with MCI depends on the source of information about IADL skills.

Due to the localization of MCI on a continuum between normal aging and the onset of dementia, functional decline is already recognizable in the MCI stage, in addition to cognitive decline [25]. Indeed, cognition plays a large role in the practical skills of daily living, skills that are part of the spectrum of instrumental ADLs [25,54]. If individuals with MCI experience limitations in these instrumental ADLs, such limitations indicate the onset of the conversion to dementia [21]. Therefore, the association between the increasing cognitive deficits related to MCI and the progressive impairments in practical skills of daily living found in the analysis is not surprising.

### 4.3. Strengths and Limitations

There are limitations with regard to the sample. The primary reason for using this sample was the participation in the randomized controlled trial (RCT). If people were recruited only for participation in a survey, the sample might be composed differently. Therefore, this sample is not representative of people with MCI. Our participants had an above-average level of education, which also means that these persons may be able to live more health-consciously. Thus, the prevalence of the risk factors and comorbidities was very low and could not be considered in the analysis. Another limitation refers to the self-assessment (activities and B-ADL), which could be affected by response bias, such as social desirability.

Further, the results are limited by the digital setting. We could only reach people who were willing to be examined remotely via video conference. In addition, study participation required technical equipment, such as a laptop or a personal computer with a camera and a microphone. A stable internet connection was also needed to participate in the twelve nutritional counseling online sessions. In some regions of Germany, internet connections are still not sufficient, which presented great challenges in conducting this study. Moreover, in the elderly German population, the use of digital devices is not yet common. On the other hand, because the study was conducted remotely, it was possible to recruit a total of 270 participants from across Germany. Due to the cross-sectional design of the study, we were not able to determine the causality of our findings.

Positive aspects of the study include the gender balance of the sample and solid randomization. Moreover, the participants were properly pre-examined, whether they could be included in the MCI sample. In addition, the data showed remarkable completeness, probably due to the individual video consultation and phone interview, where the participants could not skip a question by mistake. Last but not least, we used validated measurement instruments, such as the MoCA and the Bayer-ADL scale.

## 5. Conclusions

Concerning the protective activities:First, the combination of various activities (social, mental, and physical) are associated with better cognitive performance; thus, the recommendation should include activities from different domains;Second, for people with advanced MCI, social activities could work protectively against poor cognitive performance. Efforts should be made to help people maintain their social contacts, especially for advanced MCI.

Concerning the limitations of instrumental ADLs:Third, limitations in practical skills of daily living, such as memory or orientation, can already be detected in the MCI stage;Fourth, women perform better on memory tasks, whereas men seem to have fewer problems with orientation. These differences could be attributed to a generational and socialization effect. Therefore, men and women with MCI could receive training in the areas of their gender-specific deficits to compensate for their limitations.

## Figures and Tables

**Table 1 nutrients-15-03519-t001:** Sample characteristics of individuals with MCI.

Demographics	M (SD)/*n* (%)
Age (year)	70.8 (6.9)
Sex	
female	141 (52.2)
male	129 (47.8)
Education (ISCED)	
low	7 (2.6)
medium	104 (38.5)
high	159 (58.9)
Occupation (yes)	76 (28.1)
Living alone	80 (29.6)
**Cognitive skills**	
MoCA	22.4 (1.6)
MMSE	27.5 (1.6)
**Activities of daily living**	
physical	10.9 (9.0)
mental	18.3 (14.8)
social	10.0 (8.0)
total	39.2 (22.4)

Notes. ISCED = International Standard Classification of Education; MoCA = Montreal Cognitive Assessment; MMSE = Mini-Mental-State Examination.

**Table 2 nutrients-15-03519-t002:** Limitations of practical skills in daily living in individuals with MCI.

	Sum of Practical Skills in Daily Living	General ADL Competencies	Specific Tasks	Short- and Long-Term Memory	Orientation	Cognitive Skills
none	230 (85.2)	246 (91.2)	257 (95.2)	218 (80.7)	224 (83.0)	171 (63.3)
slight	39 (14.4)	23 (8.5)	12 (4.4)	50 (18.5)	42 (15.6)	96 (35.6)
noticeable	1 (0.4)	1 (0.4)	1 (0.4)	2 (0.7)	4 (1.5)	3 (1.1)

Notes. ADL = activities of daily living; Cognitive Skills = cognitive functions important for the management of daily living. Practical skills in daily living were collected using the Bayer-ADL scale and were classified into none, slight, and noticeable limitations in accordance with Erzigkeit and Lehfeld’s [41] specifications.

## Data Availability

The data presented in this study are available upon reasonable request from the corresponding author. The data are not publicly available due to privacy.

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
