# Peer review of "Are Protective Activities and Limitations in Practical Skills of Daily Living Associated with the Cognitive Performance of People with Mild Cognitive Impairment? Baseline Results from the BrainFit-Nutrition Study"

_nutrients, 2023, doi:10.3390/nu15163519_

Round 1
Reviewer 1 Report
The study design has several strengths, including the use of a randomized controlled trial (RCT) design. However, I have identified some areas that need improvement and clarification to enhance the quality and impact of the research.
In the introduction, you briefly mention that ADL limitations may result from physical or cognitive impairments, but it is not clear how these two factors are differentiated in the study. To strengthen the rationale, please provide a clearer distinction between physical and cognitive impairments as they relate to ADL limitations in individuals with MCI. Additionally, it would be beneficial to report the patients' chronic diseases and physical health status as potential confounding factors in the study.
Regarding the analysis, I am wondering why not utilized multivariable regressions to explore further.
Please further discuss other potential limitations such as concerns about potential reporting bias.
While the research highlights the limitations of a cross-sectional study design in determining causality, the conclusions stress the potential protective effects of various activities on cognitive functions. This is confusing.
Minor editing of English language required.
Author Response
Answers to the comments
A) Review Report Form
1) Does the introduction provide sufficient background and include all relevant references?
(Can be improved)
We improved the introduction and background. We put more suitable references and removed
redundant references. We excluded following publications: Petersen et al. (2004), Peres et al. (2006).
We included following recent publications: Hou et al. [4], Gaugler et al. [6], Cipriani et al. [20],
Samtani et al. [44], Reppermund et al. [46], Nowrangi et al. [47], Zhao et al [53], Mcalister et al. [54].
2) Is the research design appropriate? (Can be improved)
The research design was structured more clearly. Also, we extended the calculations and
conducted multivariable regression analyses.
3) Are the methods adequately described? (Must be improved)
The methods were described more detailed and more comprehensibly. In addition, we conducted,
according to your advice, multivariable regressions with the bivariate significant results.
4) Are the results clearly presented? (Must be improved)
We tried to reformulate the results in order to make it more comprehensible. We added the
results of the multivariable regression analyses and revised the use of the same terminology for
better comprehensibility. At the request of the editor, we have also added non-significant results.
5) Are the conclusions supported by the results? (Must be improved)
We have checked the completeness of the results and the respective part in the discussion. In
addition, we formulated the conclusion more carefully and removed some key points that were not
very suitable. Now it should fit to the cross-sectional design.
B) Specific comments
6) In the introduction, you briefly mention that ADL limitations may result from physical or
cognitive impairments, but it is not clear how these two factors are differentiated in the
study. To strengthen the rationale, please provide a clearer distinction between physical and
cognitive impairments as they relate to ADL limitations in individuals with MCI.
Thank you for your comment. In the initial abstract, we did not properly distinguish the two
objects of our analysis. We tried to explain it better and rewrote the abstract as well as parts of the
manuscript.
On the one hand, we measured the activity level for social, mental, and physical activity. Those
activities are well known as protective factors. Here, we expected a better cognitive performance.
On the other hand, the limitations of daily living were examined using the Bayer-ADL-scale (which
comprises basic activities of daily living and instrumental activities of daily living). We investigated
whether limitations in practical skills of daily living could be associated with a lower cognitive
performance.
7) Additionally, it would be beneficial to report the patients’ chronic diseases and physical
health status as potential confounding factors in the study.
Thank you for this remark. We also collected data on health status of the participants. However,
our study population was obtained for an RCT, therefore the sample is relatively small. In addition,
due to the recruitment strategy, the study participants were well educated and the comorbidities
and risk factors were very low (most of them below 15%). We added a brief description in the
instruments as well as results section.
8) Regarding the analysis, I am wondering why not utilized multivariable regressions to explore
further.
Thank you for your helpful advice. During the manuscript process we had already considered a
multivariable analysis; however, we had discarded the idea. Following your suggestion, we have now
included them in a sensitivity analysis. We conducted multiple regressions for the bivariate
significant results. You can find the results of the multivariable analysis behind each bivariate
significant result.
9) Please further discuss other potential limitations such as concerns about potential reporting
bias.
Thank you very much for this advice. We have added further limitations as you suggested.
Because the strength and limitation have become quite long and informative, we put an extra
heading in the discussion.
10) While the research highlights the limitations of a cross-sectional study design in determining
causality, the conclusions stress the potential protective effects of various activities on
cognitive functions. This is confusing.
You are absolutely right on that point. We reformulated the conclusion so it better fits the crosssectional design.
11) Minor editing of English language required.
The whole manuscript has been critical reviewed by our scientific English language editor, Dr. Jane
Zagorski.

Reviewer 2 Report
General considerations
The manuscript titled “Limitations in Activities of Daily Living for People with Mild Cognitive Impairment: Baseline Results from the BrainFit-Nutrition Study” is an original article on the baseline results of a randomized clinical trial, the BrainFIT-Nutrition study. In this study, the authors associate the limitations of daily life (social, mental, and physical) with mild cognitive impairment, in addition to highlighting the importance of such understanding for the prevention of progressive dementia. The study is clear and well-structured.
Specific comments
● The authors must be concise, in the title and in the abstract, regarding the content that will be addressed in the manuscript. It is necessary to align the hypothesis with the objectives and make them clear in the title and abstract of the manuscript;
● Still in the title and abstract, the methodological design of the study is confusing. I suggest that authors use a checklist to align the manuscript's topics, according to the methodological outline followed in the article;
● Authors should update their study references for recent publications (last 5 years).
Author Response
Answers to the comments
A) Review Report Form
1) Does the introduction provide sufficient background and include all relevant references?
(Can be improved)
We improved the introduction and background. We put more suitable references and removed
redundant references. We excluded following publications: Petersen et al. (2004), Peres et al. (2006).
We included following recent publications: Hou et al. [4], Gaugler et al. [6], Cipriani et al. [20],
Samtani et al. [44], Reppermund et al. [46], Nowrangi et al. [47], Zhao et al [53], Mcalister et al. [54].
2) Are all the cited references relevant to the research? (Must be improved)
We put more suitable references and removed redundant references. As you can already see in
the answer to bullet point 1, we have already added recent references in the introduction section.
We also added a few current references in the discussion.
Additionally to bullet point 1, we excluded the following – older – publications: Petersen et al. (2004),
Peres et al. (2006). We included following recent publications: Hou et al. [4], Gaugler et al. [6],
Cipriani et al. [20], Samtani et al. [44], Reppermund et al. [46], Nowrangi et al. [47], Zhao et al [53],
Mcalister et al. [54].
3) Is the research design appropriate? (Must be improved)
The methods were described in more detail and more comprehensible. We added information in
the instruments section, reformulated the statistical analysis section and additionally conducted– on
advice of the other reviewer – a multivariable regression analysis with the bivariate significant
results. Further, we added the cognition-related subgroup-analysis for the second aspect, the
limitations in practical skills of daily living.
4) Are the methods adequately described? (Can be improved)
As you can find in the section above, the methods were described in more detail and more
comprehensible.
B) Specific comments
5) In this study, the authors associate the limitations of daily life (social, mental, and physical)
with mild cognitive impairment.
Thank you for the comment. In the initial abstract, we did not properly distinguish the two objects
of our analysis. We tried to explain it better and rewrote the abstract as well as parts of the
manuscript.
On the one hand, we measured the activity level for social, mental and physical activity. Those
activities are well known as protective factors. Here, we expected a better cognitive performance.
On the other hand, the limitations of daily living using the Bayer-ADL-scale (which comprises basic
activities of daily living and instrumental activities of daily living). We investigated whether
limitations in practical skills of daily living could be associated with a lower cognitive performance.
6) The authors must be concise, in the title and in the abstract, regarding the content that will
be addressed in the manuscript. It is necessary to align the hypothesis with the objectives
and make them clear in the title and abstract of the manuscript.
We reformulated the title and the abstract in order to make it more consistent for the readers. In
addition, we tried to distinguish the two objectives clearly in the whole manuscript and hope it
corresponds with your remarks now.
7) Still in the title and abstract, the methodological design of the study is confusing. I suggest
that authors use a checklist to align the manuscript’s topics, according to the methodological
outline followed in the article.
Thank you for the advice. Indeed, some phrases in the abstract and introduction concerning the
design were unclear. We reformulated the confusing sentences. Further, we also checked the entire
manuscript to ensure completeness using the STROBE Statement-checklist.
8) Authors should update their study references for recent publications (last 5 years).
Thank you for this comment. We checked the whole manuscript and updated the literature.
We excluded following – older – publications: Petersen et al. (2004), Peres et al. (2006).
We included following recent publications: Hou et al. [4], Gaugler et al. [6], Cipriani et al. [20],
Samtani et al. [44], Reppermund et al. [46], Nowrangi et al. [47], Zhao et al [53], Mcalister et al. [54].

Round 2
Reviewer 1 Report
Accept in present form